# Studying on Alloying Elements, Phases, Microstructure and Texture in FH36 Ship Plate Steel

**DOI:** 10.3390/ma16134762

**Published:** 2023-06-30

**Authors:** Dong Wang, Guanglong Li, Wei Yin, Ling Yan, Zhenmin Wang, Peng Zhang, Xiaodong Hu, Boyong Li, Wanshun Zhang

**Affiliations:** 1School of Materials & Metallurgy, University of Science & Technology Liaoning, Anshan 114051, China; 2State Key Laboratory of Metal Material for Marine Equipment & Application, Anshan 114001, China

**Keywords:** ship plate steel, equilibrium calculation, precipitated phase, texture, grain boundary

## Abstract

This study used simulation software and experiments to analyze the microstructure and texture of FH36 ship plate steel at different thicknesses and temperatures. The austenite phase transformed into ferrite phase at 830 °C and MC and M_7_C_3_ phases precipitated at 1150 °C and 543 °C, respectively. At room temperature, the microstructure at the surface and 1/4 thickness consisted of polygonal ferrite, acicular ferrite and granular bainite, while the 1/2 thickness had less acicular ferrite and granular bainite. The texture components were mainly {111}<110> and {111}<112> at all thicknesses, but {001}<110> was stronger at 1/2 thickness. The grain size decreased gradually from 1/2 thickness to the surface, and the proportion of high-angle grain boundaries was significantly lower at the surface than at 1/4 and 1/2 thickness.

## 1. Introduction

As we all know, ships sailing in rivers, seas or oceans are subjected to the impacts of strong winds, waves and changed environmental temperatures on the performance of materials. Therefore, the requirements of the strength, toughness, fatigue property, weldability and corrosion resistance of structural materials for ship hulls are higher than those for general use. Therefore, a series of AH, EH, DH and FH high-strength or ultra-high-strength ship plate steel have been developed in order to meet the requirements mentioned above [1,2,3,4,5,6]. This kind of ship plate steel is usually made of low carbon steel containing Si and Mn, with added Mn, Cr, Ni, Nb, Ti and V and other microalloying elements, and the high strength and toughness of the steel can be obtained by the solid solution, fine grains or precipitation of carbide, such MC and M_7_C_3_ (M was the microalloying elements mentioned above) strengthening [7].

For example, Di et al. reported that the strength and low temperature impact toughness of FH36 steel plate can be significantly improved by adjusting the Nb element content in FH36 steel from 0.025% to 0.035%, and adding the 0.38% Ni element content, where the “%” symbol represents here the weight percentage [8] and the same as below. Wu et al. studied the effects of different contents of C, Si, Mn, P, S and Al_s_ (acid soluble aluminum) elements on the mechanical properties of the DH36 steel plate. The results showed that no matter whether the content of Si and P was high or low, the different Mn contents had no significant effect on the yield strength. However, when C content was lower or S and Al_s_ content was higher, the yield strength decreased significantly with the increase in Mn content. When C content was more than 0.125%, S content was less than 0.125% or Al_s_ content was less than 0.026% and the yield strength increased significantly with the increase in Mn content. In addition, when Si content was less or Al_s_ content was more, the impact of energy also decreased significantly with the increase in Mn content [9]. Peng et al. found that the precipitates played a role in precipitation strengthening and inhibited the growth of austenite phase grains, thereby improving the strength of the steel plate by adding Nb and Ti elements into DH40 steel [10]. Li et al. found that the nucleation of the acicular ferrite phase was induced by adding Mg and Zr elements in FH40 steel. As a result, the ferrite phase grains were refined and the volume fraction of pearlite was reduced. It was worth noting that the effect of adding Mg alone was better than that of adding Zr alone or adding an Mg–Zr composite [11]. Zhou et al. found that the ferrite phase grains were also refined by adding V, V–N and V–Nb elements in C–Mn steel under the conditions of hot rolling and normalizing and effectively improved the strength of the steel. By comparison, steel can obtain excellent strength and toughness matching through the addition of V–N elements [12]. 

In addition to the above-mentioned ship plate steel, the demand for F-grade ship plate steel suitable for cold sea areas had greatly increased due to the development of shipping in cold sea areas. For example, FH36 ship plate steel has been used in important parts of ships, such as the longitudinal bulkhead and th outer plate and horizontal bracket of the world’s most advanced cold sea aquaculture workboat “JOSTEIN ALBERT” in Norway.

To ensure the safe operation of ships in such cold sea conditions, FH36 ship plate steel has been studied by using fast and accurate JMatPro v7.0 and Thermo-Calc v5.0 calculation software based on thermodynamics and related databases or modules to predict the variation of microstructure, phases and elements in the phases with cooling temperature after rolling. The predicted results were compared and analyzed with the experimental results, so as to provide a design reference for stabilizing the control of the element content, phase and microstructure in ship plate steel during the production process and ensure excellent mechanical properties.

In addition, ship plate steel should also have good crack arrest properties and a small ductile–brittle transition temperature, which is strongly influenced by texture. For example, Kaneko et al. reported that the crack arrest toughness value of the steel plate at −10 °C was greater than 6000 N/mm and the ductile–brittle transition temperature was significantly lower when controlling the thermo-mechanical control process (TMCP) to form a certain proportion of specific texture components, such as <110>//X, in which X was the rolling direction [13,14]. Some research results showed that the texture intensity varied with the thickness of steel plates [15,16,17]. For example, Inagaki reported that the texture intensity increased continuously from the surface to 1/2 thickness of the low carbon steel plate [17]. Therefore, the macro-texture and micro-texture at different thicknesses of the FH36 ship plate steel were tested and analyzed in order to clarify the distribution and formation of texture in this study.

## 2. Material and Methods

### 2.1. Material and Process

The FH36 steel ingot was prepared via continuous casting, with a cross-sectional area of 250 mm × 250 mm. The actual chemical composition was listed in Table 1. 

The steel ingot was held at 1200 °C for 4–5 h. The lower heating temperature (1200 °C) would result in a decrease in the quantity of acicular ferrite and the refinement of the microstructure, which led to a reduction in the strength of the steel and improvement in its toughness.

Then, it was directly rolled along the axis of the ingot using the TMCP rolling process parameters shown in Table 2, resulting in a steel plate with dimensions of 54 mm × 200 mm × 2200 mm. The TMCP process was divided into two stages: the first stage involved rolling in the austenite recrystallization zone to refine the austenite grains, with a final rolling temperature above 830 °C; the second stage involved rolling in the austenite non-recrystallization zone to increase the ferrite nucleation sites, with a rolling temperature controlled between 790 °C and 830 °C due to the addition of the Nb element, which significantly increased the complete austenite recrystallization temperature. The plate was then cooled under controlled conditions, with accelerated cooling by water spray to approximately 450 °C, followed by air cooling. Fast cooling increased the formation of finer ferrite grains in the steel plate, thereby improving its strength and toughness. The faster the cooling rate, the more beneficial for the strength and toughness of the steel.

### 2.2. Methods

The variation of microstructure, phases and element content in the steel plate with cooling temperature after rolling was predicted by using JMatPro and thermo-Calc software, a general steel module and a TCFE6 (TCS Steel/Fe-Alloys Database) iron-based database, respectively.

After the TMCP process, the structure, microstructure and performance at the surface and 1/4 and 1/2 thickness of the steel plate were different due to the difference in deformation and cooling rate. For this reason, samples for metallographic microscope (OM), scanning electron microscope (SEM), transmission electron microscope (TEM) and X-ray diffraction (XRD) were cut at the surface, 1/4 and 1/2 thickness of the steel plate via wire cutting method. The sampling location and samples dimensions are shown in Figure 1, where RD represents the rolling direction, TD represents the transverse direction and ND represents the normal direction. OM and SEM samples were ground with 400–1000 water sandpaper and polished and corroded with 3.5% nitric acid and 97.5% ethanol. Additionally, then, the microstructure of the samples was observed using the Zeiss Axio Vert. A1 (Oberkochen, Germany) inverted microscope and the Zeiss Sigma 500 field emission SEM. The elements in the tissue were analyzed using the Bruker XFlash 6L 100 (Berlin, Germany) energy dispersive spectrometer mounted to the scanning electron microscope (SEM). TEM samples were ground and thinned to about 50 μm, then punched for the low round pieces with a diameter of 3 mm and electrolytic double-sprayed at a 30 V electrolytic voltage and a 30 °C working temperature. The sprayed electrolyte was composed of 5% perchloric acid and 95% anhydrous ethanol. The type, composition and morphology of precipitates were observed via FEI Tecnai G2 F20 (Hillsboro, OR, USA) field emission TEM and energy-dispersive X-ray (EDX) spectrometer. The RD-TD sections of the XRD samples were ground using sandpaper. The diffraction data were measured and phase-analyzed using the X’ Pert (Almelo, the Netherlands) Poeder type XRD instrument and MDI Jade v6.0 software, respectively. The measuring range was 5°–90°, and the measuring time was 3 min. According to the phase analysis results, three incomplete pole figures were measured and obtained, and then the orientation distribution function (ODF) was calculated via the X’Pert texture software.

## 3. Results and Discussion

### 3.1. Variation of Microstructure, Phases and Element Content by Temperature

#### 3.1.1. Microstructure Variation

During the cooling process of steel, as the temperature decreased, the stability of austenite decreased. At a certain temperature, austenite underwent phase transformation and transformed into a ferrite, pearlite or bainite microstructure. Figure 2 shows the phases content variation with the cooling temperature after rolling in the microstructure of the steel plate using Thermo-calc software. From the figure, it can be seen that when the temperature dropped to 800 °C, the austenite (*γ*) phase began to transform into the ferrite (*α*) phase. Some of the remaining *γ* phase transformed into pearlite (P) below 590 °C. The remaining *γ* phase began to transform into bainite (B) at 540 °C, and the B transformation stopped when the rapid cooling stopped. The microstructure at room temperature contained 88% *α* phase, 11.8% B and a small amount of P.

#### 3.1.2. Phases Content Variation 

Figure 3 shows the curves of phase content variation by temperature of 400~1200 °C via JMatPro software. As the temperature decreased, in addition to the appearance of the α phase in steel, the solubility of carbon also decreased continuously. When it exceeded the limit of the steel’s solid solubility, the excess carbon precipitated and formed carbide phases, such as Fe_3_C, MC or M_7_C_3_. It can be seen from the figure that as the temperature decreased, the phases appearing in the steel plate were *γ*, MC, *α*, alloyed Fe_3_C and M_7_C_3_ phases. It can be seen from Figure 3a that there was only a single *γ* phase in steel at 1200 °C, and the *γ* phase began to transform into *α* phase at 830 °C. The transformation was completed at 666 °C, and the *α* phase content reached 99% at 400 °C. It can be seen from Figure 3b that the MC phase continued to precipitate from the *γ* phase from 1150 °C, and its content remained almost stable below 750 °C, with a final content of 0.1%. The alloyed Fe_3_C phase began to precipitate at 683 °C, and reached a maximum of 0.94% at 543 °C, and then transformed into the M_7_C_3_ phase, the content of which gradually decreased and completely disappeared at 521 °C. The M_7_C_3_ phase began to precipitate at 543 °C, and remained basically stable below 521 °C, with a final content of 0.74%.

Figure 4 shows the curves of phase content variation by temperature using Thermo-Calc software. According to the figure, with the decrease in temperature, the phases appearing in the steel plate were *γ*, *α*, MC′, MC″, alloyed Fe_3_C and M_7_C_3_. It can be seen from Figure 4a that the *γ* phase began to transform into *α* at 829 °C, the phase transition ended at 662 °C and the *α* phase content was 99.17% at 400 °C. It can be seen from Figure 4b that the formation temperatures of MC′ and MC″ phases were 1165 °C and 754 °C, respectively. Their contents were 0.057% and 0.06% at 400 °C, respectively, so the sum was 0.117%. While the MC phase content was 0.1%, the two were basically the same. The alloyed Fe_3_C phase began to precipitate at 683 °C, reached a maximum of 0.90% at 537 °C and dissolved and disappeared at 518 °C. The M_7_C_3_ phase began to precipitate at 537 °C, and the content was 0.713% at 400 °C.

By comparison, we found that the transition, formation temperature and mass fraction of the same phase calculated by JMatPro and Thermo-Calc software had little difference, indicating that the calculated results of the two software had good consistency.

#### 3.1.3. Element Content Variation in the Same Phase

The variation of the existing elements and their contents in each phase by temperature was further and more deeply studied using JMatPro software. Figure 5 shows the curves of element content variation in the Fe_3_C phase by temperature. It can be seen from the figure that the main alloying elements contained in the alloyed Fe_3_C were Fe, Mn, a small amount of C and Cr and a trace amount of V and Ni elements. Fe and V element content in the phase decreased continuously, while Mn element content increased continuously as the temperature decreased. Cr element content increased first and then decreased, reaching the maximum at 543 °C with a content of 6.4%, and C and Ni element content varied little. The Fe element content was 60%, and the Mn element content reached 28% at 520 °C.

Figure 6 shows the curves of element content variation in the *α* phase by temperature. It can be seen from the figure that in addition to Fe element, the *α* phase also contained Mn, Ni, Si, Cr, Cu and trace amounts of V, Al, P, C, Ti and Nb elements, in which the Si, V, Al and P element content first decreased and then basically remained stable, while the V element content gradually decreased with the temperature. Almost no V elements were found below 600 °C. Mn and Cr element content increased first and then decreased with the temperature, and their contents were highest at 666 °C and 700 °C, respectively. With the decrease in temperature, the Ni element content increased first and then remained stable, while the Cu element content increased first and reached the maximum at 666 °C, which was 0.10 wt.%, then remained stable and decreased below 500 °C. The C element content first increased and then decreased and was low overall, reaching the maximum at 683 °C but only of 0.0058 wt.%.

Figure 7 shows the curves of the element content variation in MC phase by temperature. It can be seen from the figure that NbC and TiC phases precipitated at high temperatures. As the temperature decreased, NbC content decreased gradually, while TiC phase content increased gradually and reached the maximum at 850 °C, then decreased gradually. NbC and TiC phases content both remained stable after reaching 700 °C. The VC phase content was low above 900 °C, and a large amount of VC phase began to form between 700 °C and 900 °C and did not increase significantly after that. CrC phase content began to increase at about 780 °C, reached the maximum at 683 °C, and then gradually decreased. The precipitated phases that existed were VC, NbC, TiC and a small amount of CrC phases or composite phases mentioned above at below 400 °C.

Figure 8 shows the curves of the element content variation in M_7_C_3_ phase by temperature. It can be seen from the figure that the M_7_C_3_ phase included Mn_7_C_3_, Fe_7_C_3_, Cr_7_C_3_, Ni_7_C_3_, V_7_C_3_ or the composite phases mentioned above, in which the Ni_7_C_3_ and V_7_C_3_ phase content was low. The Cr_7_C_3_ phase content reached the highest level at 543 °C, which was 17 wt.% and subsequently reaching the lowest level at 521 °C, i.e., 13 wt.% and then increased gradually. The highest content was Fe_7_C_3_ phase at 543 °C, followed by Mn_7_C_3_ phase. Fe_7_C_3_ phase content decreased and Mn_7_C_3_ increased as the temperature decreased. M_7_C_3_ phases included the Mn_7_C_3_, Cr_7_C_3_ and trace amounts of V_7_C_3_ and Ni_7_C_3_ phases below 400 °C.

#### 3.1.4. Element Content Variation in Different Phases

Figure 9 shows the curves of Mn, Cr, Ni, Nb, Ti and V element content variation in different phases with the temperature. As can be seen from Figure 9, at high temperatures, all the elements mentioned above were dissolved in the *γ* phase, and during subsequent cooling processes, they transfer from the *γ* phase to other phases. It can be seen from Figure 9a that when the *γ* phase began to transform to the *α* phase, the Mn element with the most content except the Fe element also began to transfer from the *γ* phase into the *α* phase in large quantities. When the *α* phase began to transform to alloyed Fe_3_C and Mn_7_C_3_ phases, part of the Mn element in the *α* phase was subsequently transferred into the alloyed Fe_3_C and Mn_7_C_3_ phases.

As can be seen from Figure 9b, the Cr element can exist in each phase. When the *γ* phase began to transform into the *α* phase, the Cr element also began to transfer from the *γ* phase into the *α* phase and then transfer from the *α* into the Fe_3_C and Cr_7_C_3_ phases after the formation of the Fe_3_C and Cr_7_C_3_ phases. As can be seen from Figure 9c, when the *γ* phase began to transform into the *α* phase, the Ni element also began to transfer from the *γ* phase into the *α* phase. When the *α* phase began to transform into a trace amount of the alloyed Fe_3_C and Ni_7_C_3_ phase, a trace amount of the Ni element was also transferred from the *α* phase into the alloyed Fe_3_C and Ni_7_C_3_ phases. As can be seen from Figure 9d, a large amount of the *γ* phase was transformed into the NbC phase, starting from 1152 °C, during which a large amount of the Nb element was also transferred from the *γ* phase to the NbC phase. Then, starting from 830 °C, a trace amount of the Nb element in the *γ* phase was transferred into the *α* phase during the transformation of the remaining trace amount of the *γ* phase to the *α* phase. Starting at 810 °C, almost all the trace amounts of the Nb element in the *α* phase were transferred into the NbC phase. From the comparison of Figure 9d,e, it can be seen that the transferring process of the Ti element was basically similar to that of the Nb element.

It can be seen from Figure 9f that in the process of the *γ* phase successively transforming to the VC, *α* and alloyed Fe_3_C phases, the V element was also transferred from the *γ* phase to the VC, *α* and alloyed Fe_3_C phases. As the temperature continued to decrease, the V element in the *α* phase was transferred into the VC phase in large quantities. When the temperature dropped to 665~682 °C, the V element in the *α* phase temporarily stopped transferring into the VC phase, and part of the V element was transferred from VC phase into the alloyed Fe_3_C phase. As the temperature continued to decrease, the V element in the *α*, alloyed Fe_3_C and V_7_C_3_ phases continued to be transferred into the VC phase.

The content distribution of alloyed elements in the *α*, MC and M_7_C_3_ phases at 400 °C is shown in Table 3. From the table, it can be seen that almost all of the Ni element was dissolved in *α* phase and the remaining 0.1% of the Ni element was distributed in the Ni_7_C_3_ phase. Most of the Mn element was dissolved in the *α* phase and the remaining 29% of the Mn element was distributed in Mn_7_C_3_. Most of the Cr element was distributed in the Cr_7_C_3_ phase, and the remaining 30.8% and 0.2% of the Cr element were dissolved in the *α* phase and distributed in the CrC phase, respectively. The Ti and Nb elements were both distributed in the TiC and NbC phases. The V element was almost exclusively distributed in the VC phase, and the remaining 0.8% and 0.6% of the V element were distributed in the V_7_C_3_ phase and dissolved in the *α* phase, respectively.

### 3.2. Observation of Microstructure and Phases and Element Analysis

#### 3.2.1. Observation of Microstructure

Figure 10 shows the OM images of the steel plate at the surface, ¼ thickness and ½ thickness for RD-TD section. It can be seen from the images that the microstructure was mainly composed of polygonal ferrite (P*α*), acicular ferrite (Aα) and granular bainite (GB) at the surface, in which there was P*α*, a small amount of A*α* and GB at ¼ thickness and P*α* and a small amount of pearlite (P) and GB at ½ thickness. The pearlite was formed due to the smaller deformation and slower cooling at the ½ thickness of the plate, which allowed sufficient time for the diffusion of alloy and carbon elements into the high-density region of dislocations in the deformation zone. 

Because of the high cooling rate and large deformation, the Pα phase formed at the surface was more uniform and finer than that at ¼ and ½ thickness, which improved the toughness, and the Aα phase and bainite were more than that at ¼ thickness. 

The bainite content measured via Image Pro Plus was 12.87%, which was consistent with the 11.8% calculated via Thermo-calc software (See Section 3.1.1).

Liu and Zhou et al. reported that when the microstructure of low carbon microalloy ship plate steel was composed of P*α*, A*α* and a small amount of bainite, and the grain size was less than 7 μm, the ductile–brittle transition temperature could be significantly reduced and the toughness and crack arrest properties could be improved [18,19]. The microstructure of the steel plate in this study contained more P*α* and A*α* phases and less bainite, which should result in better toughness and crack arrest performance.

Figure 11 shows the OM images of the steel plate at the surface, ¼ thickness and ½ thickness for the ND-RD section. Because the rolling deformation gradually increased from ½ thickness to the surface of the steel plate, the grains were gradually refined and elongated along the rolling direction.

Figure 12 shows the SEM images at the surface, ¼ thickness and ½ thickness of the steel plate. The white convex phase in the figure was mainly bainite, and the relatively concave phase was the *α* phase. It can be seen from Figure 12a,d that the bainite formed at the surface was relatively coarse and widely distributed. It can be seen from Figure 12b,e that there were more GB formed at ¼ thickness and the distribution was relatively dense, almost only existing at the original *γ* phase grain boundary. Figure 12c,f shows that there was only a small amount of GB at ½ thickness.

#### 3.2.2. Observation of Precipitated Phases

It can be seen from Table 3 that the precipitated phases in the steel plate contained the NbC, TiC and VC phases below 400 °C. Figure 13 shows the TEM of the two kinds of precipitates and the EDX of elements by plane scanning images. The rectangular precipitate indicated by the arrows in Figure 13a corresponded to the TiC and VC composite precipitate. The size of the precipitate was about 70 × 30 nm. From the EDX image for the Ti element, it can be seen that the substrate contained almost no Ti element, and almost all Ti element participated in the formation of the TiC precipitate, which was consistent with the calculation results in Figure 9e. The precipitates indicated by the arrow in Figure 13g might be NbC, which were approximately 15 nm in diameter and were much smaller than the TiC and VC composite precipitates in size.

#### 3.2.3. Analysis of Element Content in Microstructure

Figure 14 shows the location in the SEM image and element content distribution via line scanning in the microstructure. It can be seen from the figure that the distribution of the Mn, Cr and Ni element content in the *α* phase and bainite in the structure had little difference, while that of the C element content was significantly different. The C element content in bainite was higher than that in the *α* phase, which in bainite varied significantly from left to right, and carbides may precipitate at high C element content.

Figure 15 shows the SEM image and elements content distribution via plane scanning in the microstructure of the steel plate. According to the figure, the C element content in bainite was higher than that in the *α* phase, the distribution of C element content was not uniform, and carbon-rich regions were formed locally, which was consistent with the result of Figure 14. The C element content in coarse bainite (See left blue square areas) was higher than that in fine bainite (See right blue square areas) in Figure 15b, while the Nb, Ti, V and Cr element contents were distributed uniformly, and there was no obvious difference in the content distribution.

### 3.3. Orientation and Macro-Texture Analysis

#### 3.3.1. Orientation Analysis

Figure 16 shows the XRD diffraction patterns at different locations. It can be seen from the figure that the positions of the diffraction peaks at different locations were basically the same, and the 2*θ* diffraction angles of the three peaks were about 44.93°, 65.22° and 82.51°, corresponding to (110), (200) and (211) crystal planes, respectively. The substrate had a mainly body-centered cubic structure. The diffraction intensity for the (110) plane at the surface and 1/4 thickness was significantly higher than that at 1/2 thickness. Li et al. reported that the intensity for the (110) plane was related to the rolling temperature, and the intensity was higher as a result of higher orientation density for the (110) plane at low rolling temperatures [20]. Because the rolling temperature at the surface and 1/4 thickness was obviously lower than that at 1/2 thickness, the intensity of the former was obviously higher than that of the latter.

#### 3.3.2. ODF Analysis

Figure 17 shows the constant φ_2_ = 45° ODF sections at the surface, 1/4 thickness and 1/2 thickness of the steel plate and cubic crystal system. Comparing the section of φ_2_ = 45° in Figure 17a–c with that in Figure 17d, it can be seen that there were strong {111}<110> and {111}<112> and weak {112}<110> texture components with diffuse distribution at the surface and 1/4 and 1/2 thicknesses. The formation of {111}<110> and {111}<112> texture components may be related to the rolling deformation mechanism in the *γ* phase region and the orientation relationship during the transformation from the *γ* phase to the *α* phase during the final rolling process. We may be able to draw on the results of the study of ultra-low carbon bake hardening (BH) and IF steel. Guan et al. reported that for the ultra-low carbon high strength BH steel plate, the hot rolling deformation process in the *γ* phase region was mainly realized by the cross-slip of screw dislocations and the climb of edge dislocations, while the cross-slip of the dislocations depended on the start of the {111}<110> slip system, and the higher rolling temperature allowed the dislocations to climb more easily [21]. As a result of the alternating start of the slip system, weak {110} and other texture components may be formed in the final *γ* phase structure during the transformation from the *γ* phase to the *α* phase. To satisfy the K–S orientation relationship, the {111}<110> and {111}<112> texture components were formed after the transformation. In general, these two texture components were very beneficial to the plasticity of steel plates [22]. Compared with the other two locations, the strength of the {111}<110> texture component at 1/2 thickness was weaker, while the strength of the {111}<112> texture component at the surface was stronger. Wang et al. reported that when the Ti–IF steel was rolled in the *α* phase region at the 750 °C and 800 °C final rolling temperatures, respectively, the intensity of the {111}<110> texture component formed at 800 °C was lower than that at 750 °C [23]. Accordingly, it can be considered that the weaker {111}<110> texture component at 1/2 thickness of the steel plate was related to the lower final rolling temperature in this study.

In addition, there was a strong rotational cube {001}<110> texture component at 1/2 thickness. Nancy et al. pointed out that the cooling rate had a significant effect on the texture of high-strength hot-rolled steel plate, and the {001}<110> texture component could be formed when the rate was slow [24]. So, the texture component at the 1/2 thickness of the steel plate in this study formed due to the slow cooling rate.

#### 3.3.3. Pole Figure Analysis

Figure 18 shows the 2D pole figures at the surface, 1/4 thickness and 1/2 thickness of the steel plate. By comparing the (110) pole figures at the three locations, it can be seen that there was a <110>//RD texture component at all three locations, and the texture strength was the weakest at 1/2 thickness. It can be seen from the (200) pole figures that there was a strong <200>//RD texture component at the surface and 1/2 thickness. By comparing the (211) pole figures, it can be seen that there was {112} texture component at all the three locations, and there was a strong <112>//RD texture component at the surface. By comparing the results of polar figures and ODF analysis (see Section 3.3.2), the two were in good agreement.

### 3.4. Grain Boundary, Microstructure and Subgrain Analysis

#### 3.4.1. Grain Boundary Analysis

Figure 19 shows the inverse pole figures (IPFs) at the surface, 1/4 thickness and 1/2 thickness of the steel plate. According to the results of IPFs, the equivalent grain sizes at the surface and 1/4 and 1/2 thicknesses were 2.47 μm, 3.28 μm and 3.72 μm, respectively. The grain size decreased gradually from 1/2 thickness to the surface as the cooling rate after rolling increased gradually. According to the ASTM grain size standard, the grain size grade was 13.7, 13 and 12.4, respectively, which undoubtedly belonged to an ultra-fine grain in favor of high strength. According to the figures, the proportion of high-angle grain boundary at the surface, 1/4 and 1/2 thickness of the steel plate was 74.2%, 87% and 84.2% (here the “%” symbol represents the length percentage) and the proportion of high-angle grain boundary at the surface was obviously lower than that at the 1/4 and 1/2 thicknesses. 

It was generally believed that the energy of the high-angle grain boundary was much higher than that of the low-angle grain boundary, on which most atoms deviated from the equilibrium position. When the cracks propagated to the high-angle grain boundary, they bent multiple times, passing through the high-angle grain boundary due to the irregular arrangement of atoms. As a result, a large amount of crack propagation energy was consumed, so that fracture toughness and fatigue performance was improved [25]. For example, Lambert and Gourgues pointed out that increasing the density of high-angle grain boundaries can prevent cracks from continuing to propagate or change the direction of crack propagation, thereby improving crack arrest toughness. In contrast, the low-angle grain boundary indicated almost no crack propagation [26,27]. Bhattacharjee and Dáz-fuentes found that the grains misorientation angle in the same cleavage plane must be more than 12° or 15°, so as to prevent the crack from propagating [28,29]. Because the grain misorientation angles of the steel plate in this study were all more than 10°, it was predicted that the plate had good toughness and fatigue properties.

#### 3.4.2. Subgrain Analysis

Figure 20 shows the KAM diagrams at the surface, 1/4 thickness and 1/2 thickness of the steel plate. It can be seen from the figure that the positions with high KAM values (green lines) were mainly distributed near the grains where the meta-dynamic recrystallization had not been completed. Due to the faster cooling rate, this was not conducive to recrystallization, resulting in the largest number of subgrains that were not beneficial to recrystallization at the surface, so that the distribution range of the green lines was generally wide. At 1/4 and 1/2 thickness, due to sufficient recrystallization, the number of non-recrystallized subgrains was low and the distribution range of the green lines was much narrower. In addition, because the number of subgrains that had not completed recrystallization at the surface was the largest, the proportion of high-angle grain boundaries at the surface was undoubtedly significantly small (see Section 3.4.1) [30].

#### 3.4.3. Texture Components Analysis

Figure 21 shows the distribution of texture components and their volume percent content of RD-TD cross-sections at different locations of the steel plate. It can be seen from the figure that the texture components of the steel plate mainly included S ({123}<634>), Brass ({110}<112>), Dillamore ({4411}<11118>), <111>//X, <110>//X and <001>//X texture components, where X was the rolling direction. Among these, the <110>//X texture component had the highest content, while the content of Dillamore texture component was very low at 1/4 and 1/2 thickness, so it was not marked in Figure 20b,c. The content of S and Brass texture components decreased gradually from the surface to 1/2 thickness, which was obviously related to the gradual and sufficient recrystallization process (see Section 3.4.2) The content of the <111>//X texture component at 1/4 thickness was obviously higher than that at the other two locations, while the content of the <001>//X texture component at the surface was obviously lower than that at 1/4 and 1/2 thickness. The content of the Brass texture component was low at all three locations and showed a decreasing trend from the surface to 1/2 thickness. The content of the <110>//X texture component increased from the surface to 1/2 thickness, and the content was the highest at the thickness, up to 52.1%, which was consistent with the formation of a strong {001} <110> texture component at 1/2 thickness, seen using ODF analysis (See Section 3.3.2).

## 4. Conclusions

The variation process of microstructure, precipitated phases and alloying element content of FH36 ship plate steel by temperature was simulated using JMatPro and Thermo-Calc software, and the microstructure, structure, grain boundary and texture components of the steel plates at different locations at room temperature were tested. Through analysis and comparison, we can draw the following conclusions:The software calculation results showed that the *γ* phase started to transform into the *α* phase at 830 °C, and the transformation ended at 666 °C. The MC phase started to precipitate from the *γ* phase at 1150 °C and remained almost stable at 750 °C. The alloyed Fe_3_C phase started to precipitate at 683 °C, dissolved and disappeared at 521 °C. The M_7_C_3_ phase began to precipitate at 543 °C and subsequently remained almost unchanged. At 400 °C, the steel plate consisted of 99.17% *α* phase, 0.117% MC (TiC, VC, NbC and a small amount of CrC) phase and 0.713% M_7_C_3_ (Cr_7_C_3_, Mn_7_C_3_, a small amount of V_7_C_3_ and trace Ni_7_C_3_) phase. Among the phases, the Ni element was almost completely dissolved in the *α* phase. Most of the Mn element was dissolved in the *α* phase, and the rest was distributed in the Mn_7_C_3_ phase. Most of the Cr elements were distributed in the Cr_7_C_3_ phase and dissolved in the *α* phase. The Ti, Nb and V elements were almost or predominantly distributed in the TiC, NbC and VC phases. The TiC and VC phases observed by TEM existed in the form of a composite phase with a rectangular shape, while the Nb element existed as an independent phase of NbC, whose size was much lower than that of the composite phase of TiC and VC.OM observation showed that the microstructure at the surface was mainly composed of the P*α* and A*α* phases and GB, in which the bainite content was 12.87%, which was consistent with the calculation result. The microstructure at 1/4 thickness was mainly composed of P*α*, a small amount of A*α* phase and GB, and at 1/2 thickness was composed of the P*α* phase and a small amount of pearlite and GB due to the slow cooling rate. Due to the higher cooling rate and large amount of deformation, the P*α* phase at the surface was more uniform and finer than that at 1/4 and 1/2 thickness and there was more A*α* phase and bainite than at 1/4 thickness.ODF analysis showed that there were strong {111}<110> and {111} <112> and weak {112}<110> texture components at the surface and 1/4 and 1/2 thickness of the steel plate. The formation of the {111}<uvw> texture component may be related to the rolling deformation mechanism in the *γ* phase region and the orientation relationship during the transformation from the *γ* phase to the *α* phase during the final rolling process. The formation of a weak {112}<110> texture component may be related to rolling in the *α* phase region. In addition, there was also a strong {001}<110> texture component at 1/2 thickness, which was related to a slower cooling rate. Compared with the other two locations, the weaker {111}<110> texture component at a 1/2 thickness of the steel plate was related to the lower final rolling temperature.EBSD measurements showed that the grain sizes at the surface and 1/4 and 1/2 thicknesses of the steel plate were 2.47 μm, 3.28 μm and 3.72 μm, respectively, and the grain sizes decreased gradually from 1/2 thickness to the surface, related to the gradual increase in the cooling rate after rolling. The proportions of high-angle grain boundaries at the surface and 1/4 and 1/2 thicknesses of the steel plate were 74.2%, 87% and 84.2%, respectively, and the proportion at the surface was obviously lower than that at 1/4 and 1/2 thickness, related to the larger number of subgrains that had not completed recrystallization at the surface because of the slower cooling rate.

## Figures and Tables

**Figure 1 materials-16-04762-f001:**
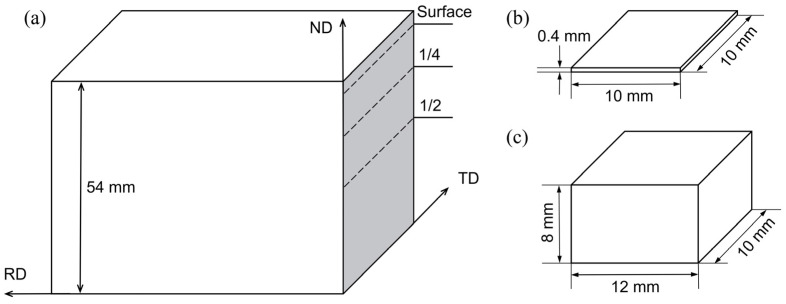
(**a**) Sampling location and sample dimensions (**b**) TEM; (**c**) OM (SEM) and XRD.

**Figure 2 materials-16-04762-f002:**
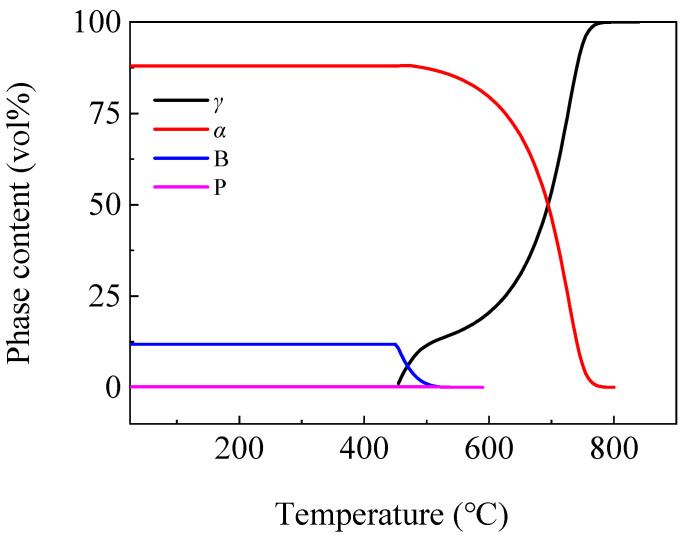
Curves of phases content variation by temperature in microstructure.

**Figure 3 materials-16-04762-f003:**
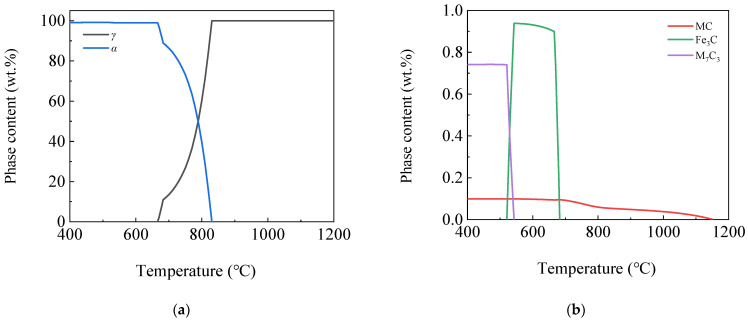
Curves of phases content variation by temperature (**a**) *γ*, *α* phases; (**b**) MC, Fe_3_C and M_7_C_3_ phases.

**Figure 4 materials-16-04762-f004:**
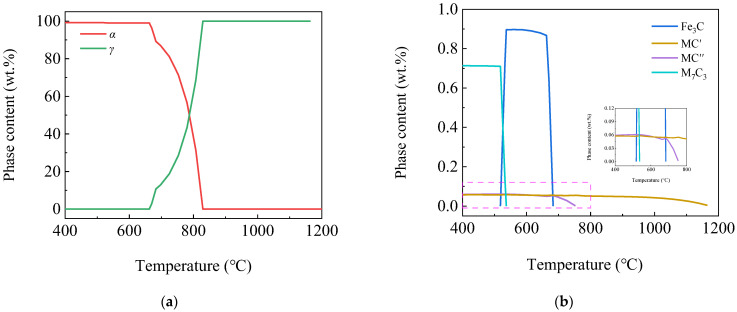
Curves of phases content variation by temperature (**a**) *γ*, *α* phases; (**b**) MC′, MC″, Fe_3_C and M_7_C_3_ phases.

**Figure 5 materials-16-04762-f005:**
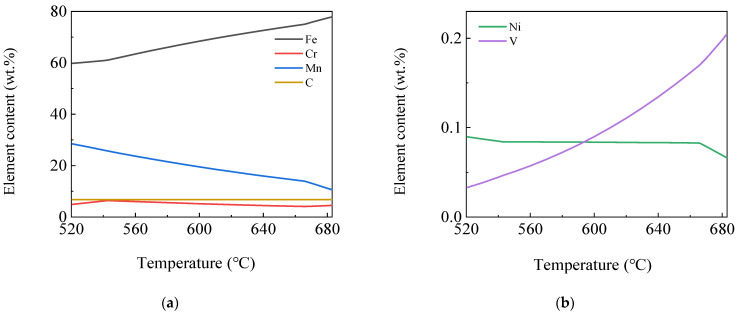
The curves of element content variation in alloyed Fe_3_C by temperature (**a**) Fe, Cr, Mn and C; (**b**) Ni and V.

**Figure 6 materials-16-04762-f006:**
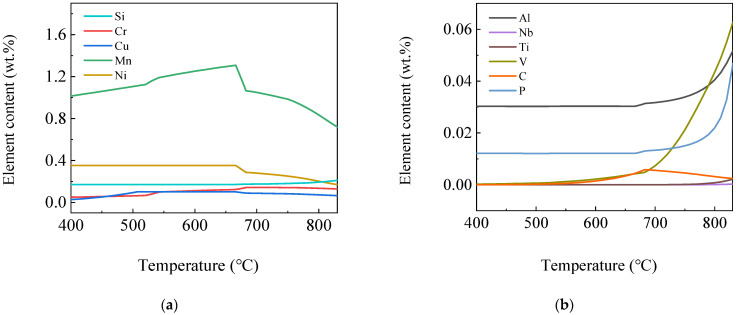
Curves of element content variation in *α* phase by temperature (**a**) Si, Cr, Cu, Mn and Ni; (**b**) Al, Nb, Ti, V, C and P.

**Figure 7 materials-16-04762-f007:**
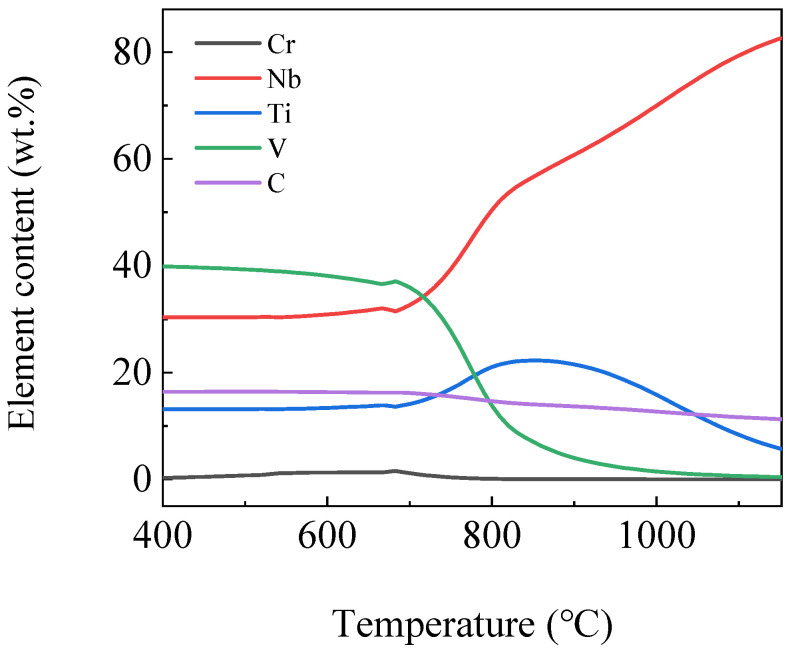
Curves of element content variation in MC phase by temperature.

**Figure 8 materials-16-04762-f008:**
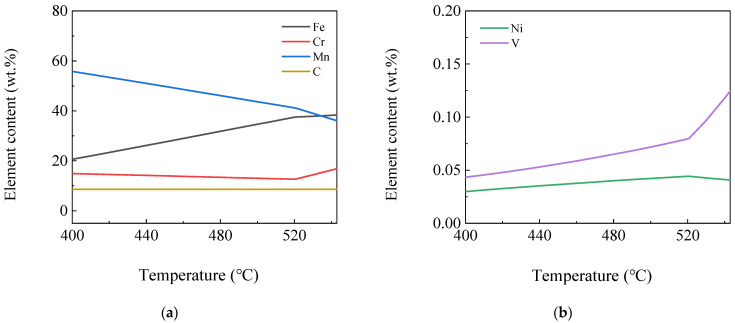
Curves of element content variation in M_7_C_3_ phase by temperature (**a**) Fe, Cr, Mn and C; (**b**) Ni and V.

**Figure 9 materials-16-04762-f009:**
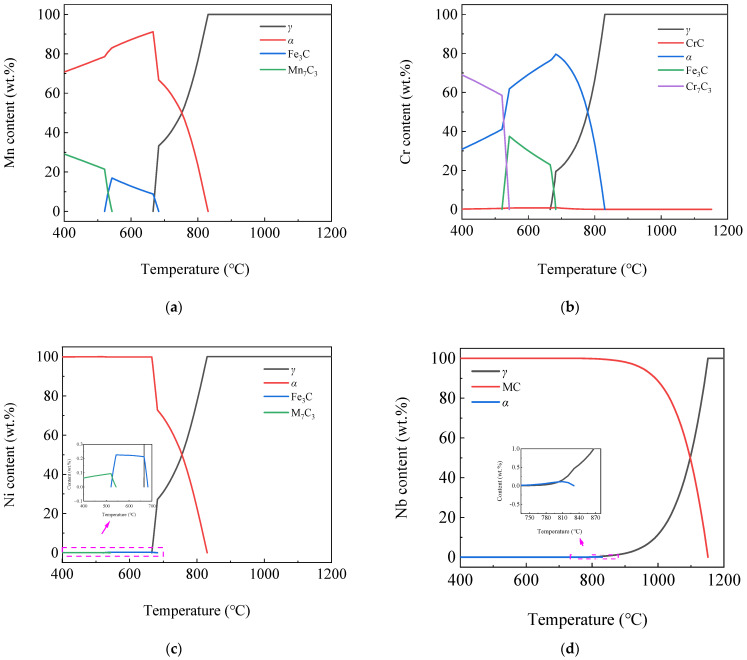
Curves of element content variation in different phases with the temperature (**a**) Mn; (**b**) Cr; (**c**) Ni; (**d**) Nb; (**e**) Ti; (**f**) V.

**Figure 10 materials-16-04762-f010:**
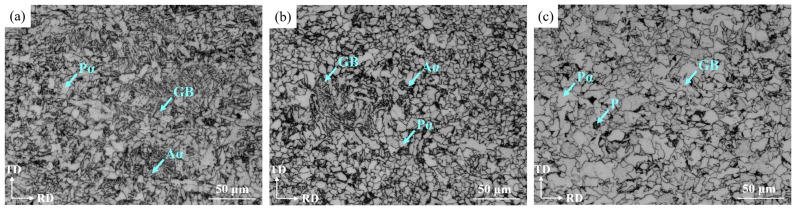
OM images for RD-TD section at different locations: (**a**) surface, (**b**) ¼ thickness and (**c**) ½ thickness.

**Figure 11 materials-16-04762-f011:**
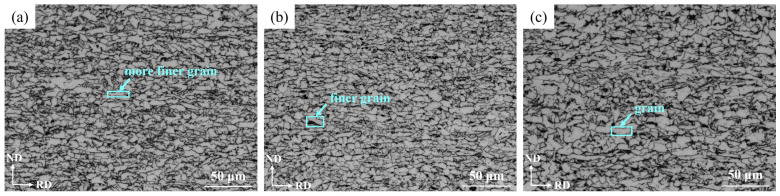
OM images for ND-RD section at different locations: (**a**) surface, (**b**) ¼ thickness and (**c**) ½ thickness.

**Figure 12 materials-16-04762-f012:**
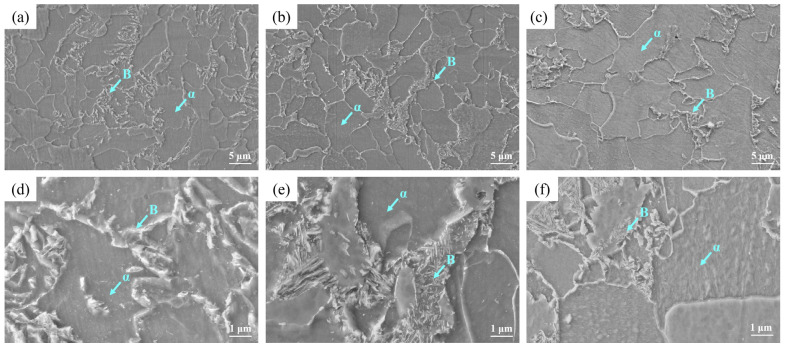
SEM images at different locations: (**a**,**d**) surface, (**b**,**e**) ¼ thickness and (**c**,**f**) ½ thickness.

**Figure 13 materials-16-04762-f013:**
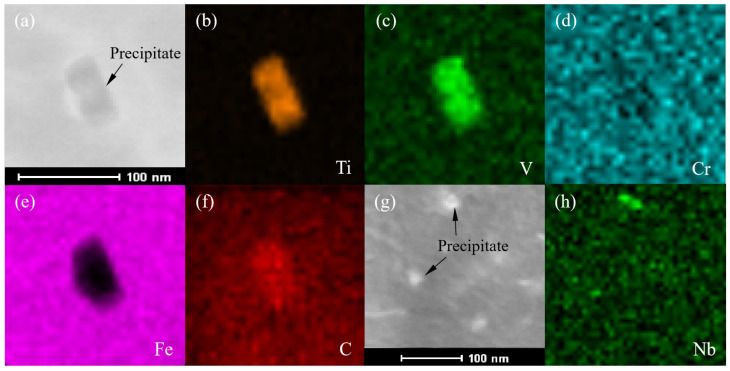
TEM of precipitated phases and the EDX of elements via plane scanning images (**a**–**f**) TiC, VC phases and contained elements; (**g**,**h**) NbC phase and contained elements.

**Figure 14 materials-16-04762-f014:**
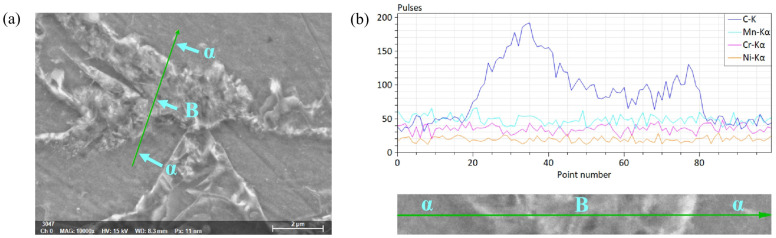
(**a**) location in SEM image and (**b**) element content distribution via line scanning in the microstructure.

**Figure 15 materials-16-04762-f015:**
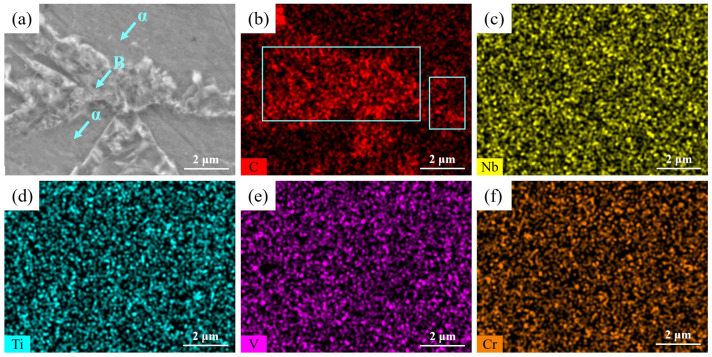
(**a**) SEM image and Element content distribution via plane scanning in the microstructure (**b**) C; (**c**) Nb; (**d**) Ti; (**e**) V; (**f**) Cr.

**Figure 16 materials-16-04762-f016:**
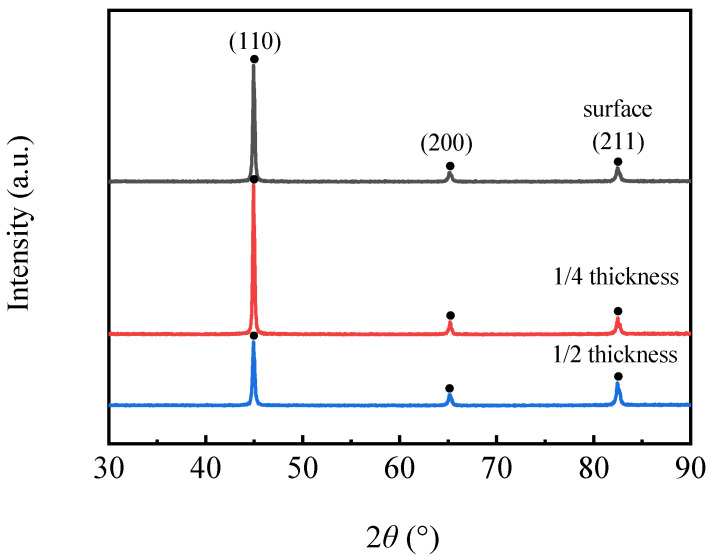
XRD diffraction patterns at different locations.

**Figure 17 materials-16-04762-f017:**
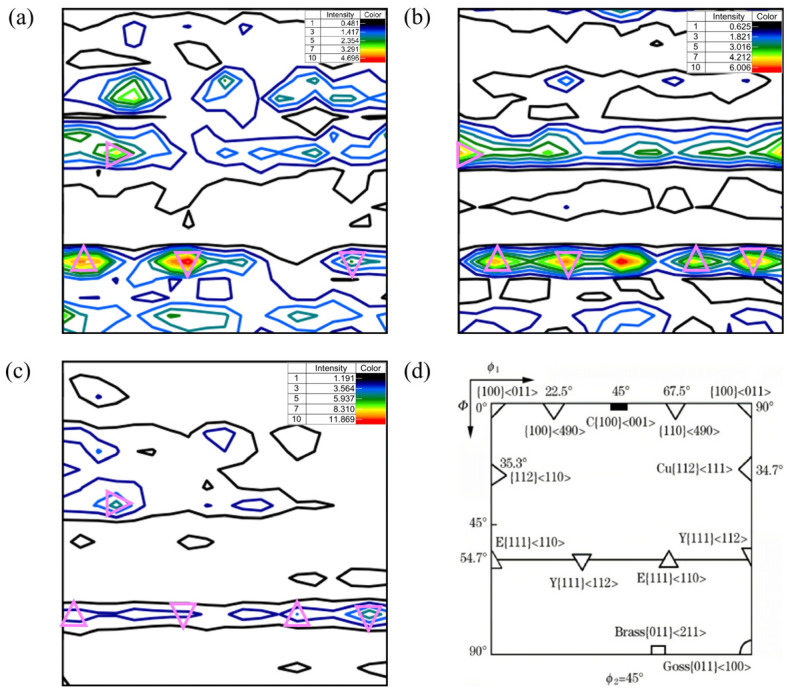
φ_2_ = 45° ODF sections at different locations: (**a**) surface, (**b**) 1/4 thickness, (**c**) 1/2 thickness and (**d**) common orientation for the cubic crystal system.

**Figure 18 materials-16-04762-f018:**
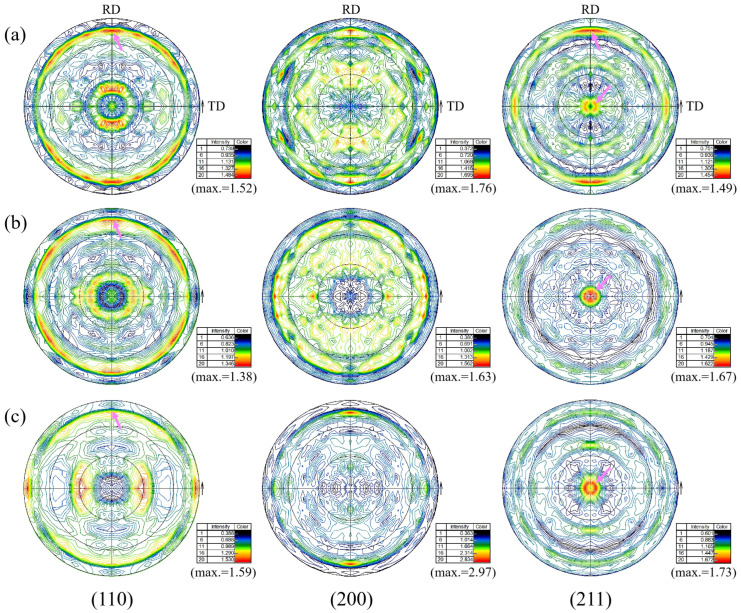
Two-dimensional pole figures at different locations: (**a**) surface, (**b**) 1/4 thickness and (**c**) 1/2 thickness.

**Figure 19 materials-16-04762-f019:**
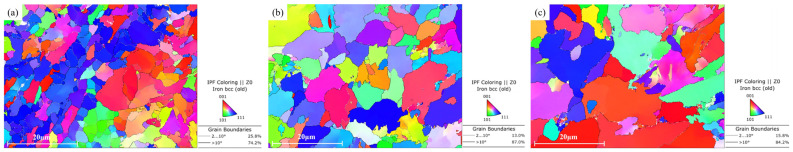
IPFs at different locations: (**a**) surface, (**b**) 1/4 thickness and (**c**) 1/2 thickness.

**Figure 20 materials-16-04762-f020:**
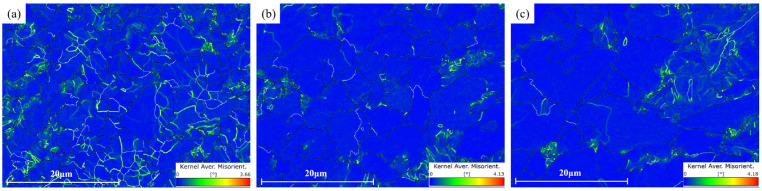
KAM diagrams at different locations: (**a**) surface, (**b**) 1/4 thickness and (**c**) 1/2 thickness.

**Figure 21 materials-16-04762-f021:**
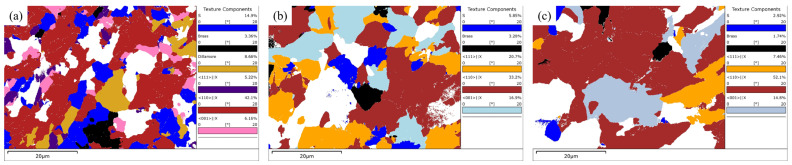
Texture components distribution images at different locations: (**a**) surface, (**b**) 1/4 thickness and (**c**) 1/2 thickness.

**Table 1 materials-16-04762-t001:** Chemical composition of FH36 steel plate (%).

Grade	Chemical Composition												
	C	Si	Mn	S	P	Nb	V	Ti	Als	Cu	Cr	Ni	Fe
FH36	0.08	0.17	1.42	0.002	0.012	0.03	0.040	0.013	0.030	0.10	0.16	0.35	Bal.

**Table 2 materials-16-04762-t002:** TMCP parameters of FH36 steel plate (°C).

Grade	The First-Stage Rolling Temperature	The Second-Stage Rolling Temperature	Final Rolling Temperature	Cooling Rate/°C·s^−1^	Final Cooling Temperature	Reddening Temperature
FH36	1050	830	790	11	450	500

**Table 3 materials-16-04762-t003:** Distribution of each alloying element content in different phases at 400 ℃ (%).

Phase	Alloying Elements					
	Ni	Mn	Cr	V	Ti	Nb
*α*	99.9	71	30.8	0.6	−	−
MC	−	−	0.2	98.6	100	100
M_7_C_3_	0.1	29	69	0.8	−	−

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
