# Peer review of "Studying on Alloying Elements, Phases, Microstructure and Texture in FH36 Ship Plate Steel"

_materials, 2023, doi:10.3390/ma16134762_

Round 1

Reviewer 1 Report

1. There are too many details in the abstract.

2. It is necessary to add a comment, what is it Als

3. Iron is not indicated in Table 1

4. In the caption to Figure 1, it is necessary to provide a transcript of the RD, TD, ND.

5. In Figure 2, pearlite and bainite are present, which are structural constituents. The title needs to be corrected or the drawing needs to be revised. It is also necessary to add the decoding of the abbreviations used.

6. It is not clear why in Figure 2 there are no phases that appear in Figure 4, what is the difference between them?

7. Section 3.1 contains too detailed description of the obtained curves. At the same time, there are practically no arguments. It would be more interesting to see explanations why there is a change in the content of one or another element or transformation.

8. When describing the curves in Figure 8, the Mn7C3, Fe7C3, Cr7C3, Ni7C3, and V7C3 phases are mentioned. It is not clear why this is so, but not about complex carbides, for example, (Fe, Mn)7C3.

9. In section 3.2.1 and in Figure 10, the designations (Pα, Aα, GB) are indicated, which are not deciphered anywhere. In addition, only GB occurs later in the text. Again, these designations appear only in the conclusions.

10. Section 3.2.2 again states that the steel contains simple carbides NbC, TiC. However, Figure 13 clearly shows that the carbide is complex. Further, in Figure 13a, there is no map for one part (white area on the left). There is no iron, and it is neither titanium, nor vanadium, nor chromium. Also, this area is clearly not shown on the carbon map.

11. The map in Figure 13b is not convincing (does not match the image).

12. The caption to figure 17 does not indicate what is shown in figure e. It is also necessary to leave only the map for the angle of 45 since it is stated. In addition, in this form, the drawing is too small.

Author Response

Dear reviewer,
Thank you so much for your reviews. Our responses are as follows:
1.There are too many details in the abstract.

Please see the sentences modified in the abstract on page 1 of the revised 
version.

2.It is necessary to add a comment, what is it Als.

Please see the sentences modified at the bottom of page 1 in the revised 
version.

3.Iron is not indicated in Table 1.

Please see Table 1 modified on page 2 of the revised version.

4.In the caption to Figure 1, it is necessary to provide a transcript of the RD, TD, ND.

Please see the sentences modified at the bottom of page 3 in the revised 
version.

5.In Figure 2, pearlite and bainite are present, which are structural constituents. The title needs to be corrected or the drawing needs to be revised. It is also necessary to add the decoding of the abbreviations used.

Please see the sentences modified in the middle of page 4 in the revised 
version.

6.It is not clear why in Figure 2 there are no phases that appear in Figure 4, what is the difference between them?

Because both pearlite and bainite are microstructures composed of multiple 
phases, including ferrite, MC', MC", Fe3C, and M7C3, Thermo-Calc 
software represents pearlite and bainite content changes with temperature in 
Figure 2. In Figure 4, Thermo-Calc software represents the content changes 
of ferrite, MC', MC", Fe3C, and M7C3 phases with temperature.

7.Section 3.1 contains too detailed description of the obtained curves. At the same time, there are practically no arguments. It would be more interesting to see explanations why there is a change in the content of one or another element or transformation.

Please see the sentences modified in the top and at the bottom of page 4, and 
the sentences modified at the top of page 5 in the revised version.

8.When describing the curves in Figure 8, the Mn7C3, Fe7C3, Cr7C3, Ni7C3, and V7C3 phases are mentioned. It is not clear why this is so, but not about complex carbides, for example, (Fe, Mn)7C3.

Please see the sentencess modified on pages 7 and 8 of the revised version. 

9.In section 3.2.1 and in Figure 10, the designations (Pα, Aα, GB) are indicated, which are not deciphered anywhere. In addition, only GB occurs later in the text. Again, these designations appear only in the conclusions.

Please see the sentences modified at the top of page 11 in the revised version.

10.Section 3.2.2 again states that the steel contains simple carbides NbC, TiC. However, Figure 13 clearly shows that the carbide is complex. Further, in Figure 13a, there is no map for one part (white area on the left). There is no iron, and it is neither titanium, nor vanadium, nor chromium. Also, this area is clearly not shown on the carbon map.

The MC carbide is theoretically a simple carbide, but in reality, it may form 
complex carbides in addition to forming simple carbide. Please see Figure 13 
modified on page 13 of the revised version and its caption.

11.The map in Figure 13b is not convincing (does not match the image).

Please see the sentences modified on page 12 and Figure 13(g) and (h) 
modified on page 13 of the revised version. Table 3 indicates that Nb only 
exists in the MC carbide, while SEM surface scanning of Figure 13(h) shows 
that the compound in Figure 13(g) contains a considerable amount of Nb, so 
it is inferred that the compound may be NbC.

12.The caption to figure 17 does not indicate what is shown in figure e. It is also necessary to leave only the map for the angle of 45 since it is stated. In addition, in this form, the drawing is too small.

Please refer to Figure 17 and its caption on page 16 of the revised version. 
Thank you so much for your reviews.

Reviewer 2 Report

The presented article is devoted the studying on alloying elements, phases, microstructure and texture in FH36 ship plate steel with temperature variations. Investigation was made with the help JMatMacro and ThermoCalc software. For scientific and practical verification of the results obtained as a result of computer simulation the microstructure, structure, grain boundary and texture components 467 of the steel plate at different locations at room temperature were tested. The results of computer simulation are illustrated with clear and understandable drawings. and the results of experimental verification are understandable even to an inexperienced reader. Despite the fact that the article does not carry fundamentally new hypotheses and their confirmation, the article can be published without changes, since it contains a huge amount of background information about the various temperatures of the change in the structure of ship steel used in practice. Using just such works, subsequently, theorists generalize the material obtained and build predictive characteristics in relation to new materials for functional purposes. The article is written in good English, beautifully illustrated and contains an exhaustive bibliographic list of references. I recommend this article for publication in the journal "Materials".

Author Response

Thank you so much for your reviews.

Reviewer 3 Report

The manuscript is a very detailed experimental effort in the characterization of steel.  However, it fails to meet novelty on several accounts;

a.      There is no explanation for the selection of steel composition is given.

b.      The impact of chemistry on the selection of thermomechanical process parameters is not mentioned or the basis of thermomechanical process parameters is not described.

c.       Synergistic effect of composition and thermomechanical process parameters on microstructure and texture needs to be addressed appropriately.

Author Response

Dear reviewer,
Thank you so much for your reviews. Our responses are as follows:
a.There is no explanation for the selection of steel composition is given.

Please refer to the modified sentences in the middle of page 2 in the 
revised version.

b.The impact of chemistry on the selection of thermomechanical process parameters is not mentioned or the basis of thermomechanical process parameters is not described.

Please refer to the modified sentences at the top of page 3 in the revised 
version.

c.Synergistic effect of composition and thermomechanical process parameters on microstructure and texture needs to be addressed appropriately.

Please refer to the modified sentences on the bottom of page 2, in the middle 
of page 11 and marked sentences of page 15 in the revised version.

Thank you so much for your reviews.

Round 2

Reviewer 1 Report

no further comments

Author Response

Thank you so much for your review.

Reviewer 3 Report

Thank you for the revisions.

Author Response

Thank you so much for your review